**Data Availability Statement:** All relevant data are within the paper.

**Funding:** The author(s) received no specific funding for this work.

# Neonatal hypothermia and adherence to World Health Organisation thermal care guidelines among newborns at Moi Teaching and Referral Hospital, Kenya

**Winstone Mokaya Nyandiko**[1,2], **Paul Kiptoon**[1], **Florence Ajaya Lubuya**[1] *

**1** Department of Child Health and Paediatrics, Moi University College of Health Science, Eldoret, Kenya,
**2** Academic Model Providing Access to Healthcare, Eldoret, Kenya

* flubuya@gmail.com

## Abstract

Neonatal hypothermia is a great concern with near epidemic levels globally. In Kenya, its prevalence is as high as 87% with limited local data on the associated factors such as adherence to warm chain guidelines as recommended by the World Health Organisation (WHO) is limited. This study aimed to determine the prevalence of hypothermia and level of adherence to the WHO thermal care guidelines among newborns admitted at Moi Teaching and Referral Hospital (MTRH). It adopted a prospective study design of following up neonates for the first 24 hours of admission to the MTRH newborn unit. Thermometry, interview of mothers and observation of thermal care practices was done. Descriptive and inferential statistical techniques were adopted. Specifically, Pearson's chi-square test of associations between predictors of neonatal hypothermia and management outcomes was conducted with their corresponding risk estimates at 95% confidence interval. Among the 372 participants, 64.5% (n = 240) were born at MTRH, 47.6% (177) were preterm and 53.2% (198) had birth weights below 2500 grams. Admission hypothermia was noted among 73.7% (274) and 13% (49) died on the first day of admission. Only 7.8% (29) newborns accessed optimal thermal care. Prematurity, day one mortality and adherence to the warm chain were significantly (p<0.001) associated with admission hypothermia. Inappropriate thermal appliance, inadequate clothing and late breastfeeding significantly increased the risk of neonatal hypothermia. Absence of admission hypothermia increased the likelihood of neonatal survival more than twenty-fold (AOR = 20.91, 95% CI: 2.15–153.62). Three out four neonates enrolled had admission hypothermia which was significantly associated with prematurity, lack of adherence to warm chain and increased risk of neonatal mortality on the first day of life. There was low adherence to the WHO thermal care guidelines. This should be optimized among preterm neonates to improve likelihood of survival.

**Competing interests:** The authors have declared that no competing interests exist

## Introduction

Newborns lose heat through conduction, radiation, convection or evaporation [1]. The World Health Organisation (WHO) defines neonatal hypothermia as an axillary temperature below 36.5˚C (97.7˚F) among newborns aged below 28 days [2]. It is stratified as either mild (36˚C-36.4˚C), moderate (32˚C-35.9˚C) and severe hypothermia (<32˚C) with the severity scale carrying prognostic implications [3]. Neonatal hypothermia has been associated with a number of risks factors such as physiological and behavioural characteristics of the neonate and caregivers as well external factors such as the environmental conditions [4]. In Spain, hypothermia was associated with very low birth weight (VLBW) among infants [5]. A similar finding was also reported in a study conducted in the neonatal intensive care units (NICUs) in Iran [6]. In Kenya, the normal neonatal birthweight in Kenya ranges between 2500g to 3999g, with a normal gestational age of more than 37 completed weeks [7].

Previous studies conducted within the sub-Saharan Africa region indicate sub-optimal thermal care practices, inadequate thermal education among providers and community level determinants of neonatal hypothermia [8,9]. Hypothermia has been reported as a major contributor to neonatal mortality due to its occurrence as a comorbidity alongside other causes of newborn deaths such as sepsis, birth asphyxia and prematurity which are prevalent in Kenya [10–12]. To reduce hypothermia-associated neonatal deaths, the World Health Organisation proposed a ten-step 'warm chain' guideline [13] that include: warm delivery rooms with temperatures between 25˚C to 28˚C at the birthplace; immediate drying before the delivery of the placenta using pre-warmed towels; skin to skin contact (SSC) for mother and baby that is among principles of kangaroo mother care (KMC); early breastfeeding (within one hour) or at least on the first day of life; delayed weighing and bathing for at least 6 hours and 24 hours respectively; appropriate clothing and bedding (*with at least 3 layers of dry and absorbent material*); rooming-in by keeping mother and baby together; warm transport- use of warm wrap, external heat source and skin to skin contact; warm resuscitation (*using appropriate appliances during resuscitation*) and Continued thermal care training for parents, caretakers and health workers. This study therefore aimed at determining the prevalence of neonatal hypothermia, its associated factors and the level of adherence to the WHO thermal care guidelines among newborns admitted at Moi Teaching and Referral Hospital (MTRH).

## Materials and methods

This was a prospective study following newborns for the first 24 hours of admission to the Relay Mother and Baby Hospital's (RMBH) newborn unit (NBU) of Moi Teaching and Referral Hospital (MTRH) in Eldoret-Kenya between July and December 2016. At the time of data collection, the newborn unit had challenges offering appropriate thermal care equipment with only seven functional incubators in place. There was also a strain on space limiting the number of newborns considered for Kangaroo mother care. The study systematically recruited 372 neonates admitted to the NBU within the first hour of admission on their first day of life. Data on warm chain management was collected by independent observers who were trained research assistants. Other than observing, they also reviewed medical records and interviewed the newborn's parent to collect sociodemographic data. All the collected data was cross-checked by the study's investigators for consistency and accuracy.

Low reading axillary thermometers (32˚C to 42˚C) and ambient air thermometers were sourced from a supplier with prior accreditation by the hospital. Eligible neonates were enrolled after completion of the admission procedure. After administering an informed consent to the mothers, neonatal axillary temperatures were taken and recorded in a pretested questionnaire where maternal sociodemographic, clinical and thermal care related details were

collected. A checklist of the available equipment, ambient temperature and observed aspects of thermal care practice was filled. Serial temperatures were taken at the 1st, 3rd, 6th, 12th and 24th hour or at the point of last contact with the newborn on day one of admission. Critical temperature values were reported to the care team and the medical records were reviewed and updated appropriately.

Descriptive statistics techniques including measures of central tendency (means, medians, frequencies and corresponding proportions) were used to describe the study participants. Inferential statistics involving Pearson's chi-square tests of association, Risk and Odds Ratio were used to draw associations among the predictor and outcome variables. Ethical approval to carry out the study was obtained from the Institutional Research and Ethics Committee (IREC) of Moi University (Approval number: 0001484) and the MTRH management. Written parental consents were obtained from either the mothers or fathers prior to enrolment of their newborns into the study as was required by the ethics committee.

## Results

### Neonatal characteristics

A total of 372 neonates with an average gestational age of 35.4 weeks (±3.9) were enrolled into the study. Among them, 57.3% (n = 213) were males while 47.6% (n = 177) were preterm. Day one mortality was observed among 13.2% (n = 49) of the participants (Table 1).

### Proportion of neonatal hypothermia

Nearly three quarters 73.7% (n = 274) of the neonates were hypothermic while only 7.8% (n = 29) accessed optimal thermal care within the 1st hour of admission. More than half (68%; n = 252) of the neonates had recurrent episodes of hypothermia on the first day of admission (Table 1). Among the neonates enrolled, majority of them had moderate hypothermia (46%) with one-tenth recording severe hypothermia (Fig 1).

Nevertheless, a steady decline in the prevalence of hypothermia was noted during the initial 24 hours of admission. Serial thermometry revealed hypothermia proportions of 55.6% (n = 207), 44.9% (n = 167), 39.8% (n = 148), 34.9% (n = 130) and 23.4% (n = 87) at the 3rd, 6th, 12th, 18th and 24th hours respectively (Fig 2). Less than one-fifth of the neonates were not assessed between the 6th and 24th hour because they were out of the newborn unit due to specialized investigations or they had been released back to their mothers in the postnatal ward.

### Adherence to individual warm chain steps by the 1st hour of admission

Provision of warm delivery rooms and newborn units at MTRH was assessed over the study period with median ambient temperatures of 24.5˚C (IQR: 22˚C, 27˚C) {n = 38}, 20.13˚C (19˚C, 22˚C) {n = 32} and 25.4˚C (24˚C, 28˚C) {n = 62} recorded at the labor rooms, operating theatres and newborn units respectively. Almost half (43.8%; n = 163) of the mothers delivered in warm rooms irrespective of the place of birth, while majority of the neonates (81.7%; n = 304) were wiped and wrapped immediately. Warm transport was observed among 243 (65.3%) neonates at admission with 82% (n = 305) of them dressed in three layers of dry, warm and absorbent clothing. However, there was notable absence of caps and stockings predominantly among the preterm babies. This lowered the aggregate score for appropriate clothing to 48.9% (n = 182) as only 53% (n = 197) neonates wore a cap and stockings. Baths were delayed among 73.7% (n = 274), while the provision of appropriate thermal resuscitation appliances was noted among 63.7% (n = 237). About two fifths (40.3%; n = 150) of the mothers had not received any thermal care education. Rooming in (4.6%; n = 17), skin to skin contact

**Table 1. Neonatal demographic and clinical profile.**

| Neonatal Characteristic | n (%)/Mean (SD) or Median (IQR) |
|---|---|
| Gestational age (weeks) Mean (SD) | 35.4 (±3.9) |
| Range (Min-max) | 26–41 |
| **Gender (n%)** | |
| Male | 213 (57.3%) |
| Female | 159 (42.7%) |
| **Place of birth (n%)** | |
| MTRH | 240 (64.5%) |
| Non MTRH | 132 (35.5%) |
| **Mode of Delivery** | |
| Operative | 100 (26.9%) |
| Non-operative | 272(73.1%) |
| **Maturity by NBMS** | |
| Term | 195 (52.4%) |
| Preterm | 177 (47.6%) |
| **Maturity by birth weight** | |
| <2500g | 198 (53.2%) |
| > = 2500g | 174 (46.8%) |
| **Diagnosis (Underlying conditions)** | |
| Prematurity | 128 (34.4%) |
| Birth asphyxia | 135 (36.3%) |
| Surgical/congenital disease | 43 (11.6%) |
| Macrosomia | 6 (1.6%) |
| Presumed sepsis | 60(16.2%) |
| **Day one outcomes** | |
| Admission hypothermia | 274 (73.7%) |
| Hypothermia recurrence | 254 (68.3%) |
| Day one mortality | 49(13.2%) |

Values represent frequency of the neonates with corresponding (%) or mean ± standard deviation; Gestational age was determined using the New Ballard's Maturity score (NBMS).

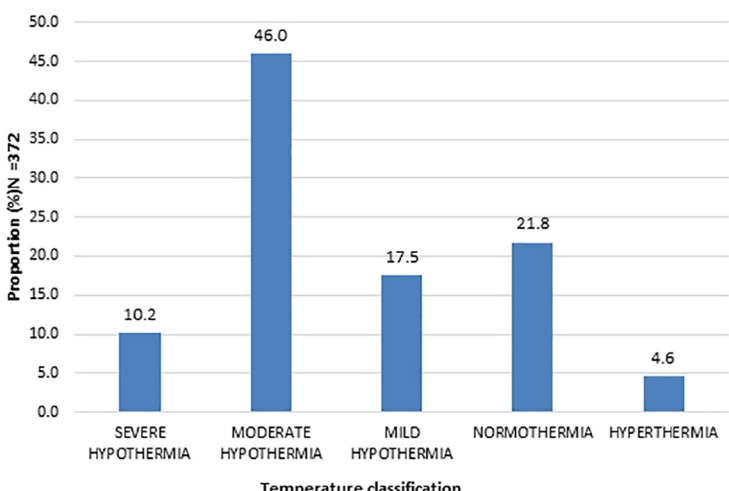

**Fig 1. Spectrum of neonatal temperatures at the 1st hour of admission.**

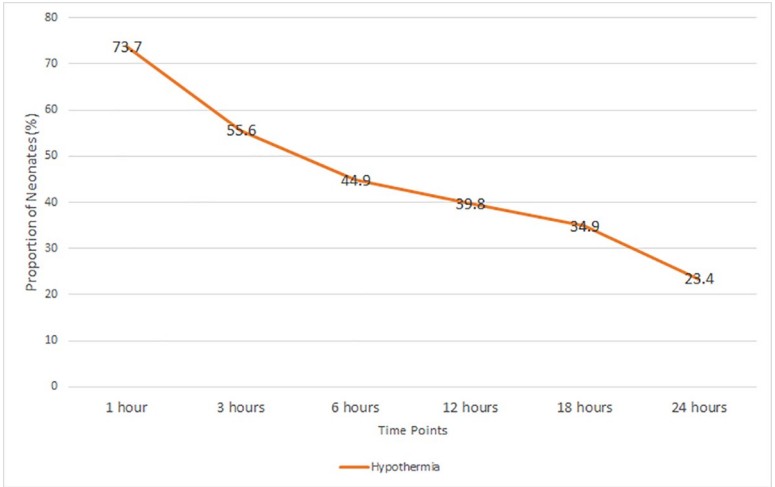

**Fig 2. Neonatal hypothermia trend during the first 24 hours of admission.**

(9.4%; n = 35) and early breastfeeding (12.5%; n = 46) were among the least adhered to steps besides the near universal immediate weighing (Table 2).

## Overall adherence to warm chain steps feasible post admission

About a half (51.6%; n = 192) of the neonates had two to three steps optimally adhered to. Only (5.1% n = 19) had optimum adherence to at least five of the post-admission steps (Fig 3).

## Optimal adherence' to the warm chain by the 1st hour

There was optimal adherence to the warm chain steps only among (7.8%; n = 29) of the participants.

**Table 2. Adherence to WHO thermal care guidelines by the 1st hour of admission (N = 372).**

| Warm chain step | n (%) |
|---|---|
| Warm delivery rooms/Newborn Units<br>*Mean ambient temperatures at MTRH (IQR)*<br>• *Labour wards (n = 38): 24.53°C (22–27)*<br>• *Obstetric theatres (n = 32): 20.13°C (19⁻22)*<br>• *Newborn unit (n = 62): 25.4°C (24°C-28°C)* | 163 (43.8) |
| Immediate wiping and wrapping | 304 (81.7) |
| Warm transport | 243 (65.3) |
| Delayed weighing | 22 (5.9) |
| Delayed Bathing | 274 (73.7) |
| **Appropriate clothing** | **182 (48.9)** |
| At least 3 layers | 303 (81.5) |
| Dry, warm and absorbent | 305 (82.0) |
| Cap/stockings | 197 (53.0) |
| **At least 1/3 warm chain steps under KMC** | **65 (17.5)** |
| Early breastfeeding | 46 (12.5) |
| Skin to skin care | 35 (9.4) |
| Rooming-in | 17 (4.6) |
| **Appropriate thermal resuscitation appliance** | 237 (63.7) |
| **Continued thermal care education** | 222 (59.7) |

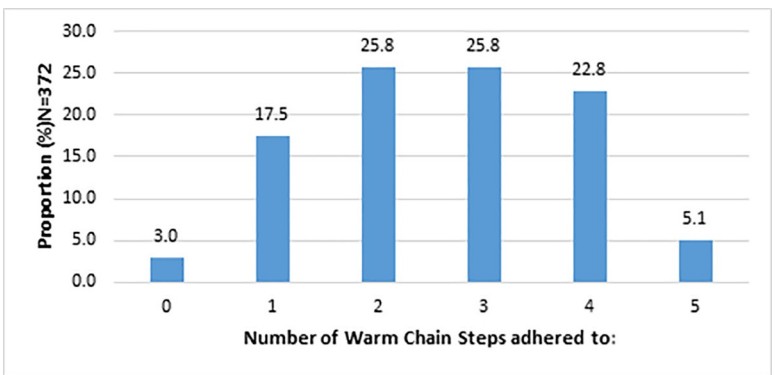

**Fig 3. Number of warm chain steps adhered to.**

## Demographic and clinical factors associated with admission hypothermia

Birthweight below 2500grams (RR = 1.58; 95% CI: 1.37, 1.82) and gestational age below 37weeks (RR = 1.62; 95% CI:1.43, 1.84) increased the risk of hypothermia at admission by 58% and 62% respectively. The risk (RR = 1.14; 95% CI: 1.01–1.28) conferred by lack of continued thermal education among the mothers was statistically significant (p-value = 0.041) as shown on Table 3.

## Association between adherence to the warm chain and admission hypothermia

Inappropriate appliances (RR = 1.50; 95% CI: 1.34, 1.67) or clothing (RR = 1.78; 95% CI: 1.54, 2.05) and sub-optimal adherence to any of the three KMC steps (RR = 1.79; 95% CI: 1.36–2.36) increased admission hypothermia risk by 50%, 78% and 79% respectively. Early bathing

**Table 3. Demographic and clinical factors associated with admission hypothermia [N = 372].**

| Variables | | Admission Hypothermia n (%) | p-value | Relative risk (95% CI) |
|---|---|---|---|---|
| **Place of birth** | Outside MTRH | 94 (71.2) | 0.427 | 0.95 (0.83–1.08) |
| | MTRH | 180 (75) | | |
| **Mode of delivery** | Operative | 75 (75) | 0.721 | 1.07 (0.73–1.6) |
| | Non-operative | 199 (73.2) | | |
| **Maturity by NBMS** | Preterm | 161 (91.0) | <0.001 | 1.62 (1.43–1.84) |
| | Term | 113 (57.9) | | |
| **Maturity-weight** | <2500grams | 176 (88.9) | <0.001 | 1.58 (1.37–1.82) |
| | ≥2500grams | 98 (56.3) | | |
| **Maternal parity** | Primiparous | 141 (76.6) | 0.198 | 1.08 (0.96–1.22) |
| | Multiparous | 133 (70.7) | | |
| **Thermal Education** | No | 119 (79.3) | 0.041 | 1.14 (1.01–1.28) |
| | Yes | 155 (69.8) | | |
| **Diagnosis** | Prematurity | 119 (93) | <0.001 | - |
| | Birth Asphyxia | 94 (69.6) | | |
| | Congenital | 33 (67.3) | | |
| | Sepsis | 28 (46.7) | | |

NBMS—New Ballard's Maturity score; MTRH -Moi Teaching and Referral Hospital.

**Table 4. Warm Chain Steps associated with admission hypothermia among neonates.**

| Variables | | Admission hypothermia n(%) | p-value | Relative risk (95% CI) |
|---|---|---|---|---|
| **Delayed bathing:** | **NO** | 75 (76.5) | 0.452 | 1.05 (0.92–1.20) |
| | **YES** | 199 (72.6) | | |
| **Appropriate appliance** | **NO** | 126 (93.3) | **<0.001** | **1.50 (1.34–1.67)** |
| | **YES** | 148 (62.4) | | |
| **Appropriate clothing** | **NO** | **178 (93.7)** | **<0.001** | **1.78 (1.54–2.05)** |
| | **YES** | **96 (52.7)** | | |
| At least 3 layers: | NO | 67 (97.1) | | 1.42 (1.30–1.55) |
| | YES | 207 (68.3) | | |
| Dry & absorbent | NO | 65 (97) | | 1.42 (1.30–1.54) |
| | YES | 209 (68.5) | | |
| Cap and Stockings | NO | 164 (93.7) | | 1.68 (1.47–1.91) |
| | YES | 110 (55.8) | | |
| **At least 1 KMC step** | **NO** | **245 (79.8)** | **<0.001** | **1.79 (1.36–2.36)** |
| | **YES** | **29 (44.6)** | | |
| Early breastfeeding | NO | 256 (78.5) | | 2.01 (1.39–2.89) |
| | YES | 18 (39.1) | | |
| Skin to skin care | NO | 267 (74.2) | | 1.27 (0.79–2.06) |
| | YES | 7 (58.3) | | |
| Rooming In | NO | 267 (75.9) | | 2.17 (1.19–3.95) |
| | YES | 7 (35) | | |

(RR = 1.05; 95% CI: 0.92–1.20) was not statistically associated (p = 0.452) with admission hypothermia (Table 4).

## Association between hypothermia, adherence to the warm chain and day-one mortality

Admission hypothermia, hypothermia recurrence within the initial 24 hours of admission and, sub-optimal adherence to warm chain guidelines were significantly associated with neonatal mortality on the first day of life (Table 5).

## Association between hypothermia, adherence to the warm chain and day one survival

Absence of hypothermia at admission (AOR = 20.907; 95% CI: 2.152–153.620) and recurrent episodes of hypothermia (AOR = 6.136; AOR = 2.152–17.496) significantly (p<0.001) increased the chances of day one survival, with a twentyfold and six-fold increased odds of survival respectively. Optimal adherence to the warm chain by the first hour, significantly improved the chances of day one survival four times.

**Table 5. Risk factors predisposing neonates to Day 1 mortality [N = 49].**

| Variables | Day one mortality | p-value |
|---|---|---|
| | n (%) | |
| **Admission hypothermia** | 48 (98%) | <0.001 |
| **Hypothermia recurrence** | 45 (91.8%) | <0.001 |
| **Sub Optimal adherence to warm chain** | 48 (98%) | <0.001 |

## Discussion

### Neonatal hypothermia prevalence

Hypothermia was prevalent among neonates admitted at MTRH with 3 out of every 4 newborns having hypothermia at admission. The high proportion of hypothermia in our study could be attributed to the frequency of the associated factors among our participants including prematurity and suboptimal adherence to the warm chain. Our prevalence results matched those from Malaysia (64.8%) [14] Nigeria (72.4%) [15] and Ethiopia 69.8% [16] NICUs. Similar challenges to provision of thermal care are expected among these hospital studies due to climatic, economic, and technological semblance hence the parallel. Lower prevalence rates of 32% [17] and 51% [18] were recorded in Brazil, while higher prevalence rates were noted in the Netherlands (93%) [19], Nepal (92.3%) [20], Zimbabwe (85%) [21] and Uganda (83%) [22]. In a local study done in a Kenyan county-referral hospital, a higher prevalence (87%) was reported [23]. The variance in hypothermia rates in these studies could be attributed to differences in temperature measuring sites, timing of thermometry, technological, economic, cultural, and ecological disparities between the study areas and the uniqueness in demographic profiles of the participants.

### Adherence to the WHO newborn thermal care guidelines

Optimal adherence to WHO thermal care guidelines among newborns was observed in less than one-tenth of all the neonates enrolled at the first hour of admission. This is parallel to findings in Malaysia where none of the NICUs evaluated practiced a complete thermal care bundle [14]. Similarly, in Nepal only 10.7% of the neonates in a community study got optimum care [24]. Reviews in Africa also cite negative and sub-standard thermal protection among the factors sustaining the epidemic of hypothermia in the region [7,25]. Provision of appropriate clothing was fairly achieved among our study population credited to the fact that the MTRH newborn unit stocked warm absorbent linen. Very few of the preterm infants weighing less than 1700 grams in our study accessed incubator care as recommended by the WHO due to an overall strain on the seven functional incubators available at the unit during the study period. Incubator sharing was rampant among the preterm infants which confirms observations that use of appropriate thermal care devices is not an optimized practice in Africa [25] unlike in European surveys which map a more technologically advanced economy [26].

### Factors associated with admission hypothermia

Gestational maturity status was significantly associated with hypothermia which concurs with findings in the Netherlands where neonates with a gestational age of less than 32 weeks had 93% hypothermia rates [27]. In Nigeria, 82.5%, (RR = 1.51,95% CI: 1.21–1.89) of the hypothermic newborns were preterm [15] with proportions similar to a study in Ethiopia (OR = 4.81;95% CI: 2.67–8.64, p-value = 0.001) and an East African meta-analysis (AOR = 4.01; 95% CI: 3.02–5.00) [28]. The risk of hypothermia among preterm neonates is innate in their physiology that limits their capacity for thermogenesis as compared to the term infant. Having a low birthweight (<2500 grams) significantly increased the risk of admission hypothermia among newborns. This compared to findings from Nepal (AOR = 4.32; 95% CI: 3.13–5.00) where newborns weighing below 2000g had a four-fold increased risk of hypothermia [29]. In the university of Iowa hospitals and clinics in the United States of America, a high rate of hypothermia (79%) was noted among very low birth weight infants [30]; whereas in Nigeria 93.3% of hypothermic babies had lower than normal weight respectively [15,31]. These findings were replicated in Ethiopia where birthweight was associated with increased

odds of hypothermia (AOR = 1.33, 95% CI:0.75–2.36) [32]. Sub optimal application of the warm chain was significantly associated with hypothermia. Comparatively, inadequate clothing among hypothermic neonates [25] doubled odds of hypothermia among neonates in Malawi who did not wear caps [33]. Increased odds of hypothermia were further noted among neonates who lacked skin-to-skin care (AOR = 2.8 95% CI: 1.46–5.66) [16] as well as those with delayed breastfeeding (AOR = 7.58, 95% CI: 3.61–15.91) [16].Similarities in setting and the congruence in challenges faced among the hospital studies could explain the parity.

### Factors associated with day one mortality and survival

There was a marked increase in the risk of day one mortality among newborns who were hypothermic at admission with increased odds of survival among non-hypothermic neonates. Similarly, a higher CFR is noted among hypothermic babies in Nigeria (37.6%) [15] vs. South Africa (16.7%). The relationship of hypothermia and death among neonates is inherent in its pathophysiologic mechanisms which involves compromises in the neonatal biological systems resulting in a cascade of vicious and deleterious events that are fatal.

## Conclusions and recommendations

This study reports sub-optimal adherence to the WHO thermal care guidelines at the MTRH newborn unit between July and December 2016 as there was less than 10% optimal adherence to warm chain steps at the first hour of admission. Prematurity and adherence to the warm chain were significantly associated with increased risk of admission hypothermia. There was a seventeen-fold increase in the risk of mortality on the first day of life among neonates who had hypothermia at admission. Although much improvement has been witnessed in the warm chain management strategies and infrastructure at MTRH, there is need to optimize the application of the warm chain guidelines quality improvement strategies addressed to the weak links identified in this study. An anticipatory approach to thermal care and priority triage of the preterm neonate admitted to newborn units should be adopted.

## Supporting information

**S1 Dataset.**
(XLSX)

## Acknowledgments

The authors would like to thank the mothers and the neonates who participated in this study. Secondly, this study could not have happened without the approval and support of Moi Teaching and Referral Hospital's administration and pediatrics department.

## Author Contributions

**Conceptualization:** Winstone Mokaya Nyandiko, Paul Kiptoon, Florence Ajaya Lubuya.

**Data curation:** Winstone Mokaya Nyandiko, Paul Kiptoon, Florence Ajaya Lubuya.

**Formal analysis:** Florence Ajaya Lubuya.

**Investigation:** Florence Ajaya Lubuya.

**Methodology:** Winstone Mokaya Nyandiko, Florence Ajaya Lubuya.

**Project administration:** Florence Ajaya Lubuya.

**Supervision:** Winstone Mokaya Nyandiko, Paul Kiptoon.

**Validation:** Florence Ajaya Lubuya.

**Writing – original draft:** Winstone Mokaya Nyandiko, Paul Kiptoon, Florence Ajaya Lubuya.

**Writing – review & editing:** Winstone Mokaya Nyandiko, Paul Kiptoon, Florence Ajaya Lubuya.

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
