## [Decision Letter · Decision Letter 0]

14 Oct 2020

PONE-D-20-15401

NEONATAL HYPOTHERMIA AND ADHERENCE TO WORLD HEALTH ORGANISATION THERMAL CARE GUIDELINES AMONG NEWBORNS AT MOI TEACHING AND REFERRAL HOSPITAL, KENYA

PLOS ONE

Dear Dr. Lubuya,

Thank you for submitting your manuscript to PLOS ONE. After careful consideration, we feel that it has merit but does not fully meet PLOS ONE’s publication criteria as it currently stands. Therefore, we invite you to submit a revised version of the manuscript that addresses the points raised during the review process.

We look forward to receiving your revised manuscript.

Kind regards,

Barbara Wilson Engelhardt, MD

Academic Editor

PLOS ONE

Journal Requirements:

a) Did participants provide their written or verbal informed consent to participate in this study?

Reviewers' comments:

Reviewer's Responses to Questions

**Comments to the Author**

1. Is the manuscript technically sound, and do the data support the conclusions?

Reviewer #1: Yes

2. Has the statistical analysis been performed appropriately and rigorously? 

Reviewer #1: Yes

3. Have the authors made all data underlying the findings in their manuscript fully available?

Reviewer #1: Yes

4. Is the manuscript presented in an intelligible fashion and written in standard English?

Reviewer #1: Yes

5. Review Comments to the Author

Reviewer #1: Nyandiko, et al. report on adherence to WHO thermal care guidelines for newborns cared for at the a hospital in Kenya. They report on 372 newborns and find that 73.9% were hypothermic upon admission. Not surprisingly, mortality rates were increased in this group. The conclusion was that adherence to WHO thermal care guidelines was extremely low.

My comments on each section of the paper.

Introduction -This section provides a nice review of previous work done in this area and summarizes WHO guidelines. I think an additional paragraph describing the setting of this hospital, problems in this setting, and the attempts at instituting the guidelines would help the reader.

Materials and Methods - Well written and clear. Ethical approval was obtained from local committee. How was the adherence to warm chain steps collected? Was an unbiased observer present or was this based on nursing memory of what happened?

Results - Demographics are listed, but it would be interesting to know what proportion of neonates with hypothermia had an underlying disease process like sepsis, etc.?

All other data and statistical tests appear to be accurate and appropriate for this type of report.

Discussion - Clear and again a good review of previous studies with integration of the data from this report to support conclusions.

Errors to correct.

page 12, line 179: "There was sub-optimal adherence to the warm chain..."

page 13, line 202: "2500g had a four-fold...."

Conclusion - Although conclusions are usually short, I do think this one could be expounded upon a little. I like the fact that the authors point to the fact that quality improvement strategies are needed to address the use of the warm chain. I think this is key and I hope a QI paper is forthcoming. Since the data from this study was collected in 2016, I hope this has taken place. Could you add a sentence or two addressing what might be implemented to prevent hypothermia?

Overall, the information in this report further contributes to the knowledge that hypothermia leads to poor outcomes and that this early step is critical to the success of caring for newborns. It provides country specific information at a large, university hospital and identifies three factors that led to increased day one mortality. It also provides a good starting point for a QI project and report that can hopefully benefit other hospitals in a similar setting.

6. PLOS authors have the option to publish the peer review history of their article (what does this mean?). If published, this will include your full peer review and any attached files.

Reviewer #1: No

---

## [Author Response · Author response to Decision Letter 0]

18 Dec 2020

Dear Reviewer and Editor,

The abstract has been summarized as recommended.

The manuscript has been edited to meet PLOS ONE's style requirements, including those for file naming.

The ethics statement has been updated including the approval number, clarification on written parental consent.

The dataset backing this study has been uploaded as supplementary information.

The grammatical errors highlighted have been cross-checked and corrected.

We have clarified additional information of the study setting both in the background and methodology sections including normal ranges for birthweight and gestational age.

The methodology has been updated to indicate that an independent observer was involved in data collection.

In the results, additional information of the proportion of neonates with underlying disease has been appended.

The discussion sections on pages 12 and 13 have been modified to correct the phrasing on optimal adherence to warm chain guidelines and the risk of birthweight to day 1 neonatal mortality.

The conclusion section has been expanded as recommended by the reviewer. The improvements that have been witnessed over time at the newborn unit has been indicated in the conclusion and recommendations section.

Lastly, the figures have been edited on the Pace website and the tables modified as per the PLOS ONE guidelines.

Thank you very much for your review.

Kind regards,

Dr. Florence Lubuya (corresponding author)

---

## [Decision Letter · Decision Letter 1]

5 Feb 2021

PONE-D-20-15401R1

NEONATAL HYPOTHERMIA AND ADHERENCE TO WORLD HEALTH ORGANISATION THERMAL CARE GUIDELINES AMONG NEWBORNS AT MOI TEACHING AND REFERRAL HOSPITAL, KENYA

PLOS ONE

Dear Dr. Lubuya,

Thank you for submitting your manuscript to PLOS ONE. After careful consideration, we feel that it has merit but does not fully meet PLOS ONE’s publication criteria as it currently stands. Therefore, we invite you to submit a revised version of the manuscript that addresses the points raised during the review process.

We look forward to receiving your revised manuscript.

Kind regards,

Orvalho Augusto, MD, MPH

Academic Editor

PLOS ONE

Journal Requirements:

Additional Editor Comments (if provided):

This is a very important report for neonatal care in Kenya and sub-Saharan Africa where such account is rare. Nyandiko, et al. report on adherence to WHO thermal care guidelines for newborns cared for at the a hospital in Kenya. They report on 372 newborns and find that 73.9% were hypothermic upon admission.

Few minor comments:

1. This is scientific report. Pleas add the aims in the introduction. The abstract has it. But it is lacking in the introduction.

2. In the statistical section please correct the "Pearson chi-square" to "Pearson's chi-squared". That is in on the abstract, the methods and the results.

3. Table 1

- Below the tables or somewhere put the meaning of the abbreviations. It needs to be a sort of clinician in English to understand and remember these abbreviations such MTRH, NBMS etc etc...

4. Table 2

- we do not need the "No(%)" column. I would recommend to remove this.

5. Table 5:

There is seems to be very small cells there. The best chi-squared here should be the Fisher's exact.

6. Table 6 is unnecessary. If the authors want keep it move to supplements.

7. Figure 2 -

- Remove the no hypothermia [we can compute it by doing 100 - Hypothermia]

- It would be nice to have below the table the amount of babies remaining like in Kapplan-Meier

8. Figure 3: I would not call it "cummulative". Strictly speaking "cummulative" plots are monotonic up i.e they grow or stay constant. Use the x-axis title "number of warn chain steps adhered to" or something similar.

Reviewers' comments:

Reviewer's Responses to Questions

**Comments to the Author**

1. If the authors have adequately addressed your comments raised in a previous round of review and you feel that this manuscript is now acceptable for publication, you may indicate that here to bypass the “Comments to the Author” section, enter your conflict of interest statement in the “Confidential to Editor” section, and submit your "Accept" recommendation.

Reviewer #1: All comments have been addressed

2. Is the manuscript technically sound, and do the data support the conclusions?

Reviewer #1: Yes

3. Has the statistical analysis been performed appropriately and rigorously? 

Reviewer #1: Yes

4. Have the authors made all data underlying the findings in their manuscript fully available?

Reviewer #1: Yes

5. Is the manuscript presented in an intelligible fashion and written in standard English?

Reviewer #1: Yes

6. Review Comments to the Author

Reviewer #1: The authors adequately addressed all of my corrections, comments, questions, and suggestions. I did note a couple of very minor grammatical errors and would recommend it is read through again to correct these and any others that might exist.

7. PLOS authors have the option to publish the peer review history of their article (what does this mean?). If published, this will include your full peer review and any attached files.

Reviewer #1: No

---

## [Author Response · Author response to Decision Letter 1]

3 Mar 2021

Dear editor,

As advised, I have made all the relevant corrections on the manuscript as indicated below:

Comment 1: This is scientific report. Please add the aims in the introduction. The abstract has it. But it is lacking in the introduction.

The study’s aims have been included in the last section of the background in the main manuscript.

Comment 2: In the statistical section please correct the "Pearson chi-square" to "Pearson's chi-square". That is in on the abstract, the methods and the results.

This correction has been made both in the abstract and methods section where Pearson's chi-square tests were mentioned.

Comment 3: Table 1, Below the tables or somewhere put the meaning of the abbreviations. It needs to be a sort of clinician in English to understand and remember these abbreviations such MTRH, NBMS etc...

The abbreviations on Table 1 have been explained in the legend section below the table. All other abbreviations used in the entire text have been fully defined the first time they are mentioned.

Comment 4: Table 2, We do not need the "No (%)" column. I would recommend to remove this.

The “No” column has been removed as advised by the reviewers.

Comment 5: Table 5, There is seems to be very small cells there. The best chi-square here should be the Fisher's exact.

Fischer’s exact test has been adopted for the results presented on table 5 (as advised by the reviewers).

Comment 6: Table 6 is unnecessary. If the authors want keep it move to supplements. 

Table 6 has been removed as per the reviewer’s guidance.

Comment 7: Figure 2 -Remove the no hypothermia [we can compute it by doing 100 - Hypothermia]

The “no hypothermia” graph has been removed from the figure 2.

Comment 8: Figure 3 - I would not call it "cummulative". Strictly speaking "cummulative" plots are monotonic up i.e they grow or stay constant. Use the x-axis title "number of warm chain steps adhered to" or something similar.

The caption on Figure 3 has been changed to “Number of warm chain steps adhered to”

---

## [Editor Report · Decision Letter 2]

8 Mar 2021

NEONATAL HYPOTHERMIA AND ADHERENCE TO WORLD HEALTH ORGANISATION THERMAL CARE GUIDELINES AMONG NEWBORNS AT MOI TEACHING AND REFERRAL HOSPITAL, KENYA

PONE-D-20-15401R2

Dear Dr. Lubuya,

We’re pleased to inform you that your manuscript has been judged scientifically suitable for publication and will be formally accepted for publication once it meets all outstanding technical requirements.

Kind regards,

Orvalho Augusto, MD, MPH

Academic Editor

PLOS ONE

Additional Editor Comments (optional):

This is important work. It documents the challenges of newborn care in one typical reference public health facility in many sub-Saharan countries.

One new minor thing remaining:

- Add below the table 1 how the gestational age was measured.
---

## [Editor Report · Acceptance letter]

15 Mar 2021

PONE-D-20-15401R2 

Neonatal hypothermia and adherence to World Health Organisation thermal care guidelines among newborns at Moi Teaching and Referral Hospital, Kenya. 

Dear Dr. Lubuya:

I'm pleased to inform you that your manuscript has been deemed suitable for publication in PLOS ONE. Congratulations! Your manuscript is now with our production department. 

Kind regards, 

on behalf of

Dr. Orvalho Augusto 

Academic Editor

PLOS ONE